# Juvenile Experience with Male Cues Triggers Cryptic Choice Mechanisms in Adult Female Redback Spiders

**DOI:** 10.3390/insects12090825

**Published:** 2021-09-14

**Authors:** Maydianne C. B. Andrade, Aiswarya Baskaran, Maria Daniela Biaggio, Maria Modanu

**Affiliations:** 1Department of Biological Sciences, University of Toronto Scarborough, Toronto, ON M1C 1A4, Canada; ais.baskaran@gmail.com (A.B.); Daniela.Biaggio@wcc.govt.nz (M.D.B.); 2Columbia Business School, Columbia University, New York, NY 10027, USA; mmodanu@gmail.com

**Keywords:** behavioural plasticity, social cues, Australian redback spider, female choice, choosiness, airborne cues, cohabitation, mate availability, sperm plugs, sexual cannibalism

## Abstract

**Simple Summary:**

Females of many species vary in their receptivity to male mating attempts. When many males are present in the habitat, the theory predicts that females should be choosy and discriminate among potential mates. When few males are available, females should mate readily with the first male who courts, and thus avoid the risk of remaining unmated. We predicted that cues perceived as juveniles that indicate male availability would affect the mating behaviour of adult females. In our first experiment, juvenile females were exposed to airborne chemicals produced by males at high or low densities. In our second experiment, we mimicked a natural situation where males or other juveniles live on the webs of females shortly before they become sexually mature, and compared this to females developing in isolation. As was consistent with our predictions, we found that the adult females changed their behaviour after exposure to cues of high male availability during development. When the females perceived many males nearby (high density airborne cues or living with males) they more often interrupted copulation, or cannibalized the males before the mating was complete as adults. In comparison, when the cues indicated low male availability, the adult females were more likely to allow the males to complete mating, and cannibalism was less common.

**Abstract:**

Female choice may be linked to population density if the expected encounter rates with potential mates affects choosiness (the energy and risk engaged to express mate preferences). Choosiness should covary with male availability, which could be assessed using the social cues available during development. We tested whether the exposure of juvenile females to cues of male density affected the mechanisms of choosiness of adult *Latrodectus hasselti* spiders in two experiments simulating natural contexts. The juvenile females were exposed to (1) volatile chemicals from two densities of adult males (airborne cues), and (2) tactile, vibrational and chemical cues from adult males or other females (cohabitation cues). As adults, the females mated readily, regardless of the treatment, but there was strong evidence for post-copulatory mechanisms of choosiness in females exposed to cues of high male availability. These included abbreviated matings (in both experiments), cannibalism of the males before the mating was complete (cohabitation), and, remarkably, a reduction in the successful placement of internal sperm plugs (cohabitation). These shifts decrease the likelihood that the first mate would monopolize paternity if the female chose to mate again. We conclude that female choosiness may impose a strong selection on males despite the high mating rates, and these effects can hinge on the cues of male availability detected by juveniles.

## 1. Introduction

Female choice patterns may show considerable variation across individuals, populations, and species, but may also shift with context [1,2,3]. Predicting the variation in the effects of female choice on male traits in nature can be challenging because mate choice arises from the interactive effects of preferences (the rank order of male traits that affect the female propensity to mate) and choosiness (the effort females are willing to expend, or the risk they will incur, to secure a mate of a particular rank, [4,5,6,7,8]), and both of these components can be plastic [1,3,9,10]. Choosiness determines mating thresholds or the slope of preference functions, and thus variation in choosiness may alter the intensity of selection on males even if the preferences are static [1,11,12,13,14]. Choosiness is expected to vary with ecological context because the trade-offs involved in forgoing one mate for others that may be encountered later can change with environmental conditions [15], predation risk [16], female mating status or condition [12,17,18,19] and demographic or social variables that indicate mate availability [20,21,22,23,24] or the variation in the quality of available mates [25,26,27]. The theory exploring the origins and dynamics of choosiness makes a range of testable predictions [21,22,28,29,30,31,32,33], but manipulative studies testing how choosiness responds to variation in natural contexts are less common than those that focus on preferences (see [3,28]). More work is urgently needed, as a recent meta-analysis suggests that the evidence for context-dependent mating behaviour, including the strength of mate choice, is weak across taxa [34], but most of the data are from studies that may not capture the most salient features of natural mating contexts (e.g., [24]).

In addition to the determination of the intensity and consistency of selection on males, variation in choosiness may be an important component of female mating strategies, particularly when female reproduction may be limited or delayed by seasonality or a scarcity of males [30,31]. One robust prediction is that choosiness should covary with the expected encounter rates with potential mates (e.g., [1,4,30,32]). When there is a risk that male availability will limit female reproduction, females that accept the first available mate will reduce the risk of remaining unmated (or of significant delays to reproduction) and may have higher fitness than choosier females (the adaptive ‘Wallflower effect’) [22,30]. Pre-copulatory choosiness (whether to mate or not with a given male) may thus vary with factors that affect this risk (e.g., [21,22,33]). In species with a sustained evolutionary history of low or unpredictable mate availability, for example, unmated females may show uniformly low pre-copulatory choosiness, typically accepting the first male that attempts to mate [28,32]. In this case, choosiness may still be plastic, but may manifest mainly through variations in post-copulatory processes [35]. Once sperm is secured through a first mating, females may be choosy about subsequent matings (‘remating’), and employ mechanisms to adjust the paternity of their offspring (i.e., cryptic or post-copulatory choice [36]). For example, mated females may have elevated remating rates if superior males are encountered later (e.g., ‘trading up’ tactics, see [22,32,37]), and they may adjust paternity to favour superior males through mechanisms like differential sperm storage or selection, or variation in copulation patterns related to paternity (reviewed in [35]).

Gauging the local availability of potential mates may thus be an important aspect of female mating behaviour in nature because of the risk of delays to mating, and the potential that pre-copulatory or cryptic choosiness might be beneficial. Social information detected by juvenile females may indicate mate availability or quality, which predicts that such information may be linked to adjustments in adult mating behaviour (e.g., [20,38,39,40]), including mechanisms of choosiness. In many invertebrates, airborne chemicals produced by conspecifics indicate information about the density, sex ratio, proximity, and developmental stage [41,42,43,44], and so could provide salient information about the local social context to developing females [45]. In addition, in some taxa, direct social information is available to juvenile females because adult males seek out, associate with, and/or guard sub-adult (final-instar) females in anticipation of mating after (or just before [46]) the female becomes an adult (‘cohabitation’ [47,48]). For example, cohabitation is common in spider species in which the first male to mate typically acquires paternity advantages [20,49,50,51,52,53,54,55]. Using these types of cues to assess mate availability may be particularly important for female web-building spiders, whose sedentary habits place limits on mate sampling, particularly in seasonal populations or those where females survive for only one mating season [56,57,58]. Spiders thus provide an opportunity to test the ways in which mate availability and choosiness are linked as a function of two types of naturally-occurring opportunities to acquire information about future mating opportunities: (1) juvenile females developing alone on their webs, exposed to airborne pheromones produced by other spiders in the population; and (2) final-instar females with cohabitants present on their webs, who are exposed to direct cues from the cohabitants (including contact pheromones, vibrational and tactile cues, and airborne pheromones). 

Here, we tested the hypothesis that cues of mate availability detected as juveniles affect the choosiness of adult females of the Australian redback spider (*Latrodectus hasselti*). We predicted that developmental exposure to the cues of males in close proximity or at high density would trigger higher levels of choosiness in adult females than if such cues were absent. We also predicted that choosiness would manifest primarily through post-copulatory mechanisms that allowed females to retain control over paternity. We did not expect high levels of pre-copulatory choosiness (e.g., mate rejection) because unmated redback females may be ‘wallflowers’ [30], for which rejecting a male outright would be costly [23]. The risk of remaining unmated may be considerable for *L. hasselti* females (up to 17% at one field site, [59,60,61]), and delays to mating reduce survivorship in this species, particularly when unmated females are food-deprived (see [62]). The population density and nearest neighbor distances in *L. hasselti* vary within and between years (e.g., [63,64], see Figure 3 in [65]), so there may be considerable variation in the encounter rates with males. The proximity or density of potential mates may be indicated by airborne pheromones, which are produced by both males and females, and have been shown to trigger developmental plasticity in males [64,66]. Additional information may be acquired directly during cohabitation, which is common during the final juvenile instar (subadult stage) in *L. hasselti* [67,68] and many other species in this genus [53,54,69,70]. Airborne cues and cues associated with cohabitation are both likely to be reliable predictors of variable adult social environments [45,65], because the female’s subadult instar lasts only about 2 weeks when food intake is high [46], and males survive for up to 1 month in nature [63]. Past work shows that *L. hasselti* females have clear preferences for prolonged courtship and for larger males, and these preferences are enforced using well-defined behaviours that affect the male’s copulation frequency, rather than the overall mating success [71,72,73,74]. In these spiders, a mating may include one or more copulations, and while one copulation is sufficient to fertilize the female’s lifetime production of eggs [61], an increased copulation frequency is positively correlated with paternity when females mate with more than one male [71]. Given the fitness costs of delaying reproduction [62], we expected females to readily accept the first male to court, regardless of the social cues of mate availability (i.e., low pre-copulatory choosiness). However, we predicted that cues of high mate availability would increase post-copulatory (‘cryptic’) processes that allow choosiness in terms of the potential adjustment of the paternity of the first mate (see [75]).

We report the results of two laboratory experiments that tested these predictions, and together allowed the assessment of the ways in which different types of cues may provide social information affecting choosiness in *L. hasselti*. In the first experiment, late-stage juvenile females were exposed to airborne chemical cues from conspecifics in two male density treatments (high or low) and when given one of two diets (high or low). This experiment simulated variation in the availability of potential mates in a population, as detected by females developing alone on their webs. In addition, the diet manipulation allowed us to probe whether female choosiness also decreased under restricted nutrient intake, as this is expected to increase the longevity cost of remaining unmated (see [62]). In the second experiment, subadult females were exposed to variations in the airborne and direct cues typical of cohabitation in three treatments (being housed alone, with juvenile females, or with adult males). In this experiment, the females received complex cues (airborne, vibrational, tactile) from conspecifics, and in the case of male cohabitants, this would simulate a situation in which a potential mate has already found a subadult female (and is present on her web). The females in this experiment were held on a constant ‘high’ diet in order to avoid diet-induced variation in the duration of the final instar, which could confound the treatment effects. As adults, females from both experiments were paired with naïve males, and pre-copulatory choosiness was measured by the variations in mating success, and the latency to the first copulation for those that mated (e.g., [76]). Post-copulatory choosiness was measured in successful matings by the frequency of copulation [71,74] and the frequency of successful sperm plug placement [77]. In *L. hasselti*, these variables determine whether the first mate will monopolize paternity, or whether females retain the capacity to cede paternity to rival males accepted as subsequent mates [71,72,73,74,77]. 

## 2. Materials and Methods

### 2.1. Natural History and Mating Behaviour

Like many *Latrodectus* spiders, *L. hasselti* (redback) females are largely sedentary throughout their juvenile instars and adulthood, whereas adult males leave their webs to search for females [75]. Once the males are on the web of an adult female, they begin courting (triggered by contact pheromones, [78,79]) with the initial (‘distal’) phase of courtship occurring entirely on the web [68,74]. After about 2 h (at 22 °C), the males make first contact with the females, and then the second (‘proximal’) phase of courtship begins, with the males moving on and off the female’s abdomen until they attempt mating after 3–5 h [68,72,80]. A complete mating involves two copulations, one with each of the male’s two intromittent organs (palps, [81]). In each copulation, the male inseminates one of the female’s two, independent sperm storage organs (spermathecae) and the two copulations are separated by an additional bout of courtship on the web (reviewed in [75]). The males maximize paternity if they inseminate both spermathecae, as sperm is apparently taken equally from both at fertilization [71,82], although a single copulation is sufficient to ensure the female’s lifelong fertility using stored sperm [61]. The copulation frequency and duration is correlated with paternity, and is at least partly under female control [67,73,74,83]. The females sometimes kill the males after a single copulation (‘premature cannibalism’), which will reduce the paternity of the first mate if the female later copulates with a rival [71,72,73]. Paternity under polyandry also depends on the placement of the sperm plugs. The tip of the palp is a defined structure with a recurved tooth at its base (apical sclerite, [84]) which breaks off during copulation in this species. It usually lodges inside the female’s genitalia [84]. If positioned deep inside the female’s reproductive tract, at the entry to the female’s spermatheca, apical sclerites prevent subsequent males from inseminating that organ (‘successful’ sperm plugs), although subsequent males can still copulate [77,81]. It is not currently clear whether variations in the plug placement are due to the males, females, or some interaction between them during mating.

Overall, then, if the first mate copulates twice and deposits a well-positioned plug in each spermatheca, he will father most of the female’s offspring even if she mates with a second male [77,81]. However, if a female accepts only one copulation, or if the sperm plugs are placed in ineffective locations, the first mate would not achieve sperm precedence if the female mates again [71]. There has been recurrent speculation, although no evidence to date, that *Latrodectus* females may be able to affect the successful placement of sperm plugs [71,75,82,83,85].

### 2.2. Spider Rearing

All of the *Latrodectus hasselti* in these experiments were from an outbred, laboratory population of *L. hasselti* derived from individuals collected in Sydney, Australia, and were reared on a reversed 12 h light/dark cycle in a temperature-controlled room (25 ± 5 °C). The spiderlings were fed *Drosophila* twice per week, and were initially reared communally (with siblings). At the 3rd–4th instar, the spiderlings were separated into individual cages. Females past their 5th instar were then fed one adult cricket (*Acheta domesticus* or *Gryllodes sigillatus*) each week, and were checked every 48 h for moulting. The subadult females (7th instar, one moult prior to sexual maturity) were identified by their developmental record and morphology (i.e., swelling in the area covering the developing genitalia, but an absence of genital openings). The males were fed *Drosophila* sp. throughout their development, but this feeding ceased at adulthood (as is typical for males in nature, [86]). The males were identified at the subadult instar (4th instar, one moult prior to sexual maturity) by the presence of enlarged palps, and were monitored every 48 h for moulting. The adult males were identified by their mature palps, which are visible to the naked eye [53,54,70,75,84]. All of the spiders were weighed prior to their use in the experiments (on an Ohaus Explorer balance accurate to 0.01 mg, Ohaus Corp. Persippany, NJ, USA).

### 2.3. Experiment 1: Airborne Cues

This experiment used a 2 × 2 design, crossing airborne cues of the male density (high or low) with female diet (high or low) while controlling for the female density. In order to control for variation among family lines, four female siblings from each of 10 unrelated dams were removed from the laboratory population when the developmental records indicated that they were in their 6th instar (one moult prior to the subadult stage). One female from each sibship was randomly assigned to one of the four treatments. The spiders were held inside individual plastic cages (4 × 4 × 5 cm, Amac Plastic Products, Petaluma, CA, USA) placed in a grid pattern on plastic trays with at least 2 cm between the cages. For each cage, two of the four vertical sides of the cage were replaced with a porous screen (fine mesh black fibreglass) This permitted the exchange of airborne chemicals but no direct or vibrational interaction between the spiders. The experimental treatments were held in two separate environmental chambers (Betatek Inc., Toronto, ON, Canada) maintained at 24 °C on a 12:12 h LD cycle. In the Low mate availability chamber, the juvenile females were held at a ratio of 1 adult male to every 10 females. In the High mate availability chamber, there were 20 adult males to every 10 juvenile females. In order to avoid possible chamber effects, the treatments were exchanged between the two chambers each week. The low diet females were fed one cricket once every two weeks, and the High diet females were fed one cricket every week.

In the Low male treatment (*n* = 20 focal females), 10 juvenile females were placed on each of two trays on separate shelves within one chamber. A single adult male was placed on each tray, and the position of the male was rotated twice per week in order to equalize the male pheromone exposure for all of the females. In the High treatment (*n* = 20 focal females), each of two trays on two shelves held 10 juvenile females and 20 adult males, with the position of the individual males within the grid of females being randomized. 

The spiders were checked daily. Dead males were removed and replaced with other randomly chosen males. The female moult dates were recorded. The focal females were removed from the chamber after their final adult moult, and were replaced with a new juvenile female until all of the focal females had matured into adults. After their final moult, the focal females were returned to the standard rearing room conditions, then placed in the mating trials outlined below.

### 2.4. Experiment 2: Cohabitation Cues

The cohabitation cue experiment used focal females in the subadult (final juvenile) instar, one of the periods when cohabitation typically occurs [53,54,70,75]. The females were removed from the population and placed into cohabitation chambers within 2 days of moulting into the subadult instar. The females were randomly assigned to one of 3 treatments: (1) adult male cohabitant (high male density treatment, *n* = 24), (2) juvenile (4th instar) female cohabitant (low male density treatment, *n* = 23), or (3) isolation (low spider density control, *n* = 27). The females remained in the same cohabitation treatment until they moulted and were sexually mature. The focal females were fed once per week throughout the experiment, but the cohabitants were not fed.

Each cohabitation-treatment female experienced a different cohabitant each week in order to ensure that any effects were due to cohabitation per se rather than the traits of any one individual cohabitant. The field work on congeners suggests that this is within the range of typical cohabitation durations for males and subadult females (average durations between 4 and 10 days, e.g., *L. revivensis* [53,70] and *L. pallidus* [54]). The cohabitants were selected from the population of all of the spiders of the appropriate sex/developmental stage that were unrelated to the focal females. The first male cohabitant was chosen randomly from all of the suitable males, and the others were chosen to ensure that the females had experience with males across the full range of male sizes in the population. Juvenile female cohabitants were chosen from among suitable individuals that were similar in size to adult males.

The cohabitation chambers were rectangular plastic boxes (Amac Plastic Products, Petaluma, CA, USA) bisected along their long axis by a sheet of porous material (non-fusible light-weight interfacing) and accessible from both ends by tight-fitting, solid lids (Figure 1). The cohabitation chambers were held in experimental rooms with 12:12 light cycle at 22–25 °C. In order to reduce stress on the spiders, facilitate habituation to the cages, and reduce the time needed for web building, we allowed older virgin adult females (>4 months post-final moult) to build webs on both sides of each cage prior to the cohabitation treatment. In nature, subadult females often live on the periphery of the webs of adult females, and readily take over abandoned webs (MCBA pers obs).

The cohabitation treatments included three phases that gave the focal females access to the full range of cues available from their cohabitants throughout their subadult instar, but that reduced the likelihood of copulation (in nature, 35% of subadult *L. hasselti* females copulated with cohabiting males just before their final moult, [46]). The experimental period started when a focal subadult female was introduced to the web on one side of a cage, and was allowed one hour to habituate. In the preliminary trials, the females added silk to the existing web during this time, explored the cage, then settled into a typical quiescent posture. A cohabitant was then introduced to the same side of the cage, and their interactions were observed for one hour (the full cues period) during which time tactile, vibratory and airborne information was available. The cohabitant was then moved to the empty web in the opposite chamber for one week during which airborne chemical and (potentially) vibrational cues could be detected by focal females (the signal cues period). At the end of the week, the cohabitant was removed, and the focal female was moved to the other side of the cage in order to allow interaction with the chemical cues in the silk deposited by the cohabitant (the silk cues period). The focal female was left for one hour to experience the silk cues without other stimuli. The experience sequence was repeated (full cues, signal cues, silk cues). This continued until the focal female moulted, at which point any cohabitants present were removed, and the females were returned to their rearing cages. 

### 2.5. Mating Trials: Airborne and Cohabitation Cue Experiments

The mating trial procedures were similar after both development experiments. After moulting to the adult stage, the females were fed one cricket per week. Within 3 weeks of their adult moult, the females were weighed and placed on a frame for 24 h to build a web. The frames consisted of two inverted U-shaped metal arches 7.5 cm high separated by 8 cm, on a plexiglas rectangular base (22 × 7.5 cm) surrounded by water (to prevent escapes), inside a larger plastic container (Rubbermaid^®^, Greensboro, NC, USA, 43.2 × 28.2 × 17.1 cm). In the airborne cues experiment, the four female siblings in each treatment were paired with unrelated males, which were themselves siblings and similar in size in order to reduce any variation in response that might be linked to the male phenotype. In the cohabitation cues experiment, the males were chosen randomly from among those naïve mature males in the laboratory population that had not been used in the experimental treatments, ensuring that they were not related to the focal female. In addition, for females that had experienced male cohabitants, we ensured that the mating trial male was not related to any of the males to which the female had previously been exposed.

The mating trials were conducted at 24 ± 2 °C during the scotophase under dim red-light illumination, as redbacks are nocturnal. The trials were recorded using low-light cameras (CCTV camera, Panasonic Canada Inc., Mississauga, ON, Canada) fitted with macro zoom lenses (Zoom 7000, 18–108 mm lens, Navitar Inc., Rochester, NY, USA). A mating trial commenced when a male was introduced to the web at the farthest point from the female, while she was quiescent (e.g., [61]). The trials ended after 8 h, when the male was dead (cannibalism is frequent), or if 2 h had passed after a copulation and the males were no longer courting [68,80]. 

For each trial, we noted the timing of the commencement of the male courtship activity, and whether it occurred throughout the trial. The behaviours associated with the production of vibrational signals on the web or on the female’s body are visible to the naked eye (e.g., [87,88]). For pairs that mated, we recorded the number of copulations (airborne and cohabitation experiments), and the occurrence and timing of cannibalism relative to copulation (cohabitation experiment). These were either observed during the trials, or were assessed post-hoc by reviewing the recordings of the trials.

### 2.6. Sperm Plug Success

The females that copulated were replaced in their rearing cages after mating, fed one cricket each week until they produced at least one viable egg sac to confirm the successful mating, and were then euthanized by hypothermia and preserved in 70% or 75% ethanol. In order to visualize sclerites that might function as sperm plugs, the females’ genitalia were dissected. Spermathecae and the attached insemination tubules were cleared (macerated) by treating them with 0.5 mL of 5% KOH for one week (see [84]), and were then photographed. For each spermatheca, we scored whether a sclerite was present in a position in which it would function as an effective plug (blocking the entrance of the spermatheca [77]). Some females were stored for up to 48 months before they were dissected (the plugs remain in place even in 100-year-old preserved female *Latrodectus,* see [75]).

### 2.7. Statistical Analysis

The sample sizes for the analyses vary because some of the females died during moulting or prior to mating, some spermathecae were damaged during the dissections, and some observations were missed during the mating trials and were not clearly visible on the recordings. 

All of the statistical analyses were performed using IBM SPSS Statistics (Version 27). The residual plots for most of the variables showed deviations from normality (confirmed by Lillifors tests, *p* < 0.05), so we analyzed the data using generalized linear models (GLM) with appropriate error distributions, including binary or ordinal multinomial logistic distributions (categorical data) and gamma distributions (skewed scalar data).

Airborne cues experiment: We examined whether density, diet, or their interaction affected mating success (GLM, binary logistic), or the latency to mate for females that mated (GLM, gamma distribution), as well as whether these treatments affected the number of copulations achieved, or the number of plugs found in the females after successful matings (GLM, ordinal multinomial logistic). 

Cohabitation experiment: We asked whether the cohabitation treatment affected mating success (GLM, binary logistic). For the females that mated, we tested for treatment effects on the latency to mate (GLM, gamma distribution), the number of copulations, the occurrence or timing of sexual cannibalism (GLM, binary logistic), and the number of effectively placed sperm plugs (GLM, ordinal multinomial logistic). For this experiment, the models examining the mating outcomes included male mass (correlated with size) as a covariate, as size can affect mating outcomes [72,81], and the size of the mating-trial males was not standardized across the treatments (unlike the airborne cues experiment).

In some cases, we examined whether the treatments differed in their categorical outcomes, but only for subsets of the trials (e.g., only males that achieved two copulations), and in these cases we used Freeman-Halton-Fisher exact tests, given the smaller sample sizes that resulted. 

## 3. Results

In the mating trials for both experiments, all of the males courted from their first moments on the web until they mated and/or were cannibalized during mating, or the trial ended, as is common for *L. hasselti* [68]. All of the courting males approached and attempted to mount the females, and once mounted, contacted the female’s external genitalia (epigynum) with their palps, usually repeatedly.

### 3.1. Airborne Cues Experiment

The mating success was high and similar across the treatments (85%, *n* = 40 females; Freeman-Halton-Fisher test, *p* = 0.74; Figure 2). In successful matings, the latency to copulation (i.e., the total courtship duration prior to the first copulation) was predicted by an interaction between the density and diet treatments (Table 1). The average courtship duration prior to accepting a mating was the longest for females reared with a high density of males on high diets, whereas the courtship duration in all of the other treatments was similar (Figure 3). 

The copulation frequency depended on the male density (Table 1). The females that mated after being reared with a high density of males were more likely to copulate only once, whereas the females reared with a low density of males were more likely to accept two or three copulations (Figure 3). There was no effect of the treatment on the number of sclerites found in effective positions in the females’ spermathecae (Table 1).

### 3.2. Cohabitation Experiment

Across the treatments, the duration of the final instar was 11.72 days on average (S.E.= 0.39, GLM; no Treatment effects: Wald χ^2^ = 1.69, df = 2, *p* = 0.430), and the females were exposed to a similar number of spiders (mean ± S.E.; 1.52 ± 0.15) in the male and juvenile female cohabitant treatments prior to their final moult (there were no treatment effects on the cohabitant number: GLM Wald χ^2^ = 2.4, df = 1, *p* = 0.118). 

The latency to the first copulation was similar across the treatments (Figure 4, Table 2), and most of the females mated regardless of the treatment (91%, *n* = 66 females, Freeman-Halton-Fisher exact test, *p* = 0.513; Figure 5).

There was variation in the copulation frequency as a function of the female’s cohabitation treatment. The focal females tended to be more likely to accept two copulations from their mate if they had been isolated as subadults (*n* = 21; 66.7%), or had cohabited with juvenile females (*n* = 20; 70.7%), than if they had cohabited with adult males (47.4%, *n* = 19; Figure 4), although these differences were not statistically significant (Table 2). However, the females were more likely to cannibalize their mates during the first copulation (‘premature cannibalism’) in the male-cohabitant treatment compared to the other two treatments (Table 2, Figure 6). While premature cannibalism by the focal females in the isolation or juvenile female-cohabitation treatments was rare (<10% of trials), the focal females from the male-cohabitation treatment were almost four times more likely to kill their mates after one copulation, making a second copulation impossible (Table 2, Figure 6). Thus, single-copulation matings in the male-cohabitant treatment were more frequently a clear result of female behaviour.

For the pairs that mated, the frequency of successful sperm plugs found at the entrance to the female’s spermatheca also depended on the cohabitation treatment of the female (Figure 7, Table 2). For example, more than one third of the mated females in the male-cohabitation treatment had complete plug failure; that is, no plugs were successfully placed at the entrance to the spermatheca (37%, *n* = 19) compared to less than half that frequency for the females that cohabited with juvenile females (15%, *n* = 20), and similarly for the isolation-control females (20%, *n* = 20, Figure 7A, Table 2). These results reflect the overall outcome of the mating, which combines the copulation frequency and plugging success. We also analysed the plugging success of males as a function of the number of copulations achieved in each mating. When the females copulated only once, the plug failure frequency was similar across the treatments (Figure 7B, Freeman-Halton-Fisher, *p* = 0.852). For the females that copulated twice, however, there was a striking effect of treatment, with the male-cohabitation females being much more likely to be missing one or more effective sperm plugs compared to the isolated females and those that had juvenile female cohabitants (Figure 7C, Freeman-Halton-Fisher exact test, *p* = 0.006). 

## 4. Discussion

We examined the effects of developmental exposure to cues of social context on the adult mating behaviour and choosiness of Australian redback spider females (*Latrodectus hasselti*). Our two experiments differed in the types of cues available, simulating two natural opportunities for exposure to cues of mate availability. In the airborne cues experiment, we examined the effects of the detection of males at a distance, and our two-by-two design probed the effects of the perceived male density and female diet. We predicted that male density would affect female choosiness, and expected particularly strong reductions in choosiness when the females were on a low nutrient diet, as this accentuates the longevity cost of remaining unmated in this species [63]. Second, we examined the effects of the complex cues (airborne, vibrational, tactile) typically available during female cohabitation with other spiders, comparing the effects of male or juvenile female cohabitants to a control in which the females were isolated with no cues from other spiders. Given the risks of delays to reproduction in *L. hasselti*, we predicted that unmated females would be unlikely to reject a courting male outright [22,30,75], but we expected that cues indicating high mate availability would increase the deployment of cryptic (post-copulatory) choosiness mechanisms [36,89]. Consistent with this, the females mated at high rates with the first male that courted them across both experiments, regardless of the types of social cues available (Figure 2 and Figure 5). Despite the high mating rates, we found three types of evidence of cryptic choosiness mechanisms that were affected by social cues. First, a female exposed to a high density of males and well-provisioned with a high nutrient diet took longer to accept a mate than the females with other combinations of male density and diet (Figure 3). Second, the females were more likely to copulate only once if they were exposed to cues indicating a high availability of potential mates (significantly so in the airborne cues experiment, Figure 2). Moreover, there was clear evidence of female control over the copulation frequency in the cohabitation cues experiment, in which the females that had experienced male cohabitants were more likely to cannibalize their mates after a single copulation compared to the females in the other treatments (Figure 6, Table 2). Third, for the first time in *Latrodectus,* to our knowledge (see [75,85]), we found experimental evidence that females can affect sperm plug placement, as successful plugs were significantly less common when the females experienced cohabitation cues from males during their subadult instar, and this was true even when controlling for the number of copulations achieved (Figure 7, Table 2). Taken together, our results support the hypothesis that the mating tactics of *L. hasselti* females have two components. First, the females are likely to accept first mates with little choosiness, regardless of the cues of mate availability. This is consistent with the predictions that the females should act to insure their fertility and minimize costly delays to reproduction, given an evolutionary history of a relatively high risk of remaining unmated (the ‘wallflower’ effect [30,31]). Second, in the presence of cues of high male availability, the females retain the capacity for choosiness by copulating only once with males and preventing the effective placement of sperm plugs, two mechanisms that can circumvent the first male sperm priority that is expected in the genus [71,75,77]. Moreover, there appear to be differences in the female responses to cues of mate availability in the habitat (airborne cues) compared to cues indicating that potential mates have located their webs (cohabitation cues). We conclude that, despite high mating rates, female-mediated post-copulatory sexual selection on males may be intense at high population densities in *Latrodectus hasselti*, and this is mediated by cues detected by juveniles. 

The nature of the cues received during development affected the response of adult *L. hasselti* females. Airborne cues, such as those that would be available to a female developing alone on a web, may indicate the availability of potential mates in the population [45]. The exposure to airborne cues alone from a relatively high density of males was sufficient to make the females more likely to copulate only once. This is similar to previous work in the congener *L. hesperus*, in which adult females that developed in screen cages in the field in close proximity to conspecifics were more likely to copulate only once with the first male introduced in laboratory trials, and were more likely to kill the first male that approached them, compared to females that developed while distant from conspecifics [23]. In the case of *L. hesperus*, however, it was unclear whether the response was to cues from males or from conspecifics in general [23]. In our experiment, as we held the female density constant, the increased cryptic choosiness is clearly due to the detection of males (or a male-determined variation in sex ratio), rather than due to cues from an overall increase in the spider population density. In our cohabitation experiment, we also saw significant effects of experiencing cues produced by males, but not other conspecifics (in this case, juvenile females). In this experiment, the females were exposed directly to cues from spiders that were present on their webs. These direct, complex cues (information across multiple modalities) would provide more information than the simple cue of airborne chemicals [90], and would also be more closely linked to actual opportunities for mating [23]. As was consistent with this, the females that cohabited with males as juveniles displayed a wider range of cryptic choosiness mechanisms, restricting first male precedence both through premature cannibalism and by blocking sperm plug placement (Figure 6 and Figure 7). 

In contrast to our results regarding the mechanisms of post-copulatory choosiness, we found little evidence that experiencing cues of a male-rich environment during development decreases the female receptivity to the first mating attempt. One possible exception was the longer latency to mating for the females that were reared on a high diet while experiencing airborne cues of a high density of males. These females accepted copulation only after ~1 h of additional courtship relative to the other treatments on average (Figure 2). As adult females control mating access in this extremely size dimorphic species, and the males were courting throughout this period, this delay is likely to reflect a shift in the female (rather than male) behaviour. In nature, it is possible that multiple competitors could be attracted to females during this period (see [91] for data on mate attraction in a congener), and thus a delay may provide females with an opportunity for less-costly simultaneous, rather than sequential, choice among potential mates [58]. However, this outcome only occurred for females that were well-fed, which were perhaps better able to accommodate risks of delayed mating due to increased provisioning (e.g., [62]). This calls into question whether such an effect would be common in nature, as previous work on *L. hasselti* shows that females are rarely as well-provisioned in the field as they are in our laboratory populations [62]. Thus, the implications of this delay are unclear, and would require additional study in nature. Nevertheless, this outcome occurred only in one treatment, and the more common pattern was for females to mate with the first male to court them. The relative rarity of outright mate rejection has been noted in past work on *L. hasselti* [68,92]. This supports the idea that there is selection for females to accept at least one copulation from their first suitor. Selection may arise from the risk of remaining unmated in nature, which at one field site ranged from 11–17% of *L. hasselti* females [61]. A model that considered how the risk of delays to mating affects choosiness concluded that a 1% chance of dying unmated was sufficient to drive 50% of females to mate with the first male they encountered [30]. Given that this risk appears to be an order of magnitude higher in *L. hasselti*, it is not surprising that rejections by female redbacks are rare, and that most females mate with the first male they encounter. 

The strongest and most consistent outcome of this study was the experimental demonstration that female development in the presence of male cues leads females to decrease the potential for their first mate to monopolize the paternity of their offspring. Previous work on *Latrodectus* argues that cryptic choice via sperm manipulation would be unlikely, given the morphology of the female’s genitalia [82]. However, sperm selection is only one of a large number of possible mechanisms of cryptic choice, which includes any process during or after mating that differentially affects the paternity of mates [36,89]. In *L. hasselti*, if the first male to mate copulates twice and places two plugs effectively, the outcome is that virtually 100% of the offspring are fathered by that male [71]. In other words, this outcome removes the potential for post-copulatory choice [71,72]. In contrast, if the female copulates only once, then one spermatheca is available, uncontested, for subsequent mates, who may accrue 50% paternity or more, and previous studies show that there is no cost to female lifetime fertility [61,71]. On the other hand, if a female allows two copulations but prevents the successful deposition of plugs, she has sperm in both of her spermathecae. This may provide ‘insurance’ against lateralized malfunction, and subsequent mates will typically accrue 50% paternity, as the sperm mix in the storage organs [71,77]. Selection for females to be able to manipulate plugs is expected if there are advantages to polyandry in such a system [36,93]. While the females in this experiment were not given the opportunity to mate again, in nature males arrive on female’s webs throughout the season [63], and many females have the opportunity mate multiply. A valuable follow-up study would examine whether the females that employ these cryptic tactics are also more likely to mate with subsequent males. 

One intriguing aspect of the current study is the evidence that male-experienced females can manipulate the successful placement of sperm plugs. Female control over plug retention or placement has been demonstrated in other spiders, but typically in species where the plug protrudes from the female’s genital opening and can be easily accessed (e.g., *Argiope keyserlingi* [94,95] and the whole-body plug of *Argiope aurantia*), where female fluids must be released to combine with male fluids to create a gelatinous plug [85,96,97], or where the female produces the plug herself [98]. In *Latrodectus*, the male’s sclerite must be placed deep within the female’s internal genitalia to be an effective plug [77], and so the mechanism that permits female control is unclear. Andrade and Macleod [75] speculated that the contractions of muscles attached to the internal fertilization duct [82] during copulation might alter the pressure in the genital tract, leading to hydrostatic resistance to successful plug placement. The testing of this hypothesis would require detailed morphological and physiological studies. 

Cryptic choice is often examined from the perspective of post-copulatory preference for males of particular phenotypes [35]. Similarly, past work on *L. hasselti* showed that females adjust their copulation frequency and cannibalism in response to variations in the phenotypic traits of courting males (size, courtship duration and competitive tactics [72,73,74,81]). This study, in contrast, shows the ways in which cryptic choice may also manifest as shifts in the capacity for choosiness. Copulation frequency and cannibalism were linked to the exposure of juvenile females to information about the costs and benefits of choosiness, rather than to male phenotypes. In most species, there is unlikely to be a solitary optimum reproductive tactic for females. Instead, as has been demonstrated extensively for males, temporal or spatial variation in ecological, social and demographic factors can affect the female strategies associated with higher fitness [99]. Given that variation or patchiness in population density has been reported within and across the seasons in a variety of spider taxa [45], these effects may also be common, although the relevant cues may vary. Our study supports the accumulating evidence that cues of demography detected by juveniles can be key to adaptive plasticity in adult mating traits, and suggests that this could be an important aspect of female reproductive tactics [23,70,100].

## Figures and Tables

**Figure 1 insects-12-00825-f001:**
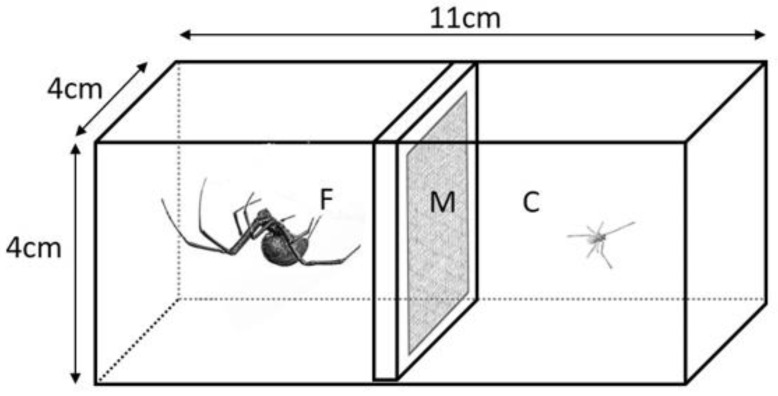
Cohabitation cages consisted of two clear plastic chambers bisected by a porous material (M, light-weight non-fusible interfacing) in a plastic frame that allowed the passage of airborne chemicals and vibrations between the chambers, but did not allow direct contact between the spiders when they were in opposite chambers. The focal subadult females (F) were held with a cohabitant (C, adult male [shown] or juvenile female), or the adjacent chamber was left empty (isolation control). The spider images (not to scale) are from photographs by Ken Jones.

**Figure 2 insects-12-00825-f002:**
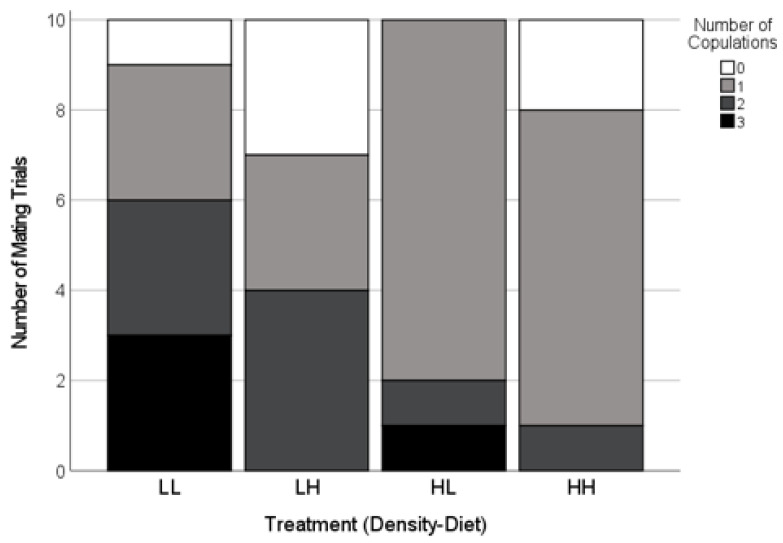
Copulation frequency of adult females paired with novel males as a function of the experimental conditions during their final two juvenile instars. The experiment crossed two male densities (H = high; L = low) with two diet levels (H = high; L = low). Very few females did not mate (white). For the trials where mating occurred, the females exposed to a low density of males (LL, LH) often copulated two or three times (dark grey and black) whereas the females exposed to a high density of males (HL, HH) more frequently copulated only once (light grey, Table 1).

**Figure 3 insects-12-00825-f003:**
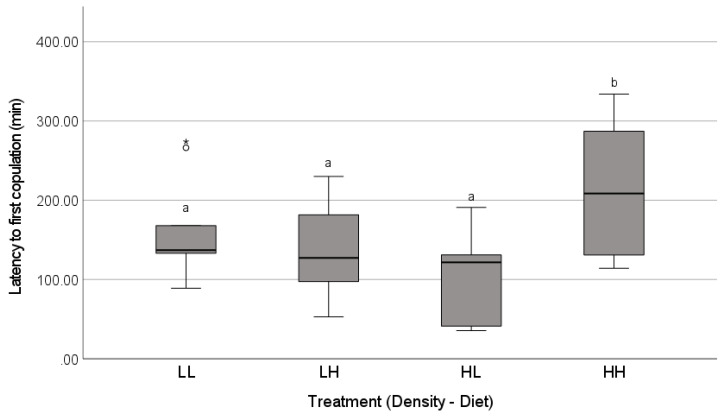
Total latency (min) prior to the first successful copulation for the females reared for their last two instars in treatments that crossed two levels of exposure to airborne chemical cues of low (L) or high (H) male density with two diet levels (low, L or high, H). The adult females were then paired with a novel male in a laboratory mating trial. Family line effects were controlled by using four females from each of 10 family lines, where one female was placed in each treatment. The boxplots show the medians (solid line), the range between the first and third quartiles (inter-quartile range, IQR, box), the maximum and minimum values excluding outliers (whiskers), a potential outlier (open circle, 1.5 − 3× IQR above the third quartile), and an extreme value (>3× IQR above the third quartile). Different letters above box plots (a,b) indicate significant differences among treatments (S = sequential Bonferroni pairwise post-hoc tests, *p* < 0.05).

**Figure 4 insects-12-00825-f004:**
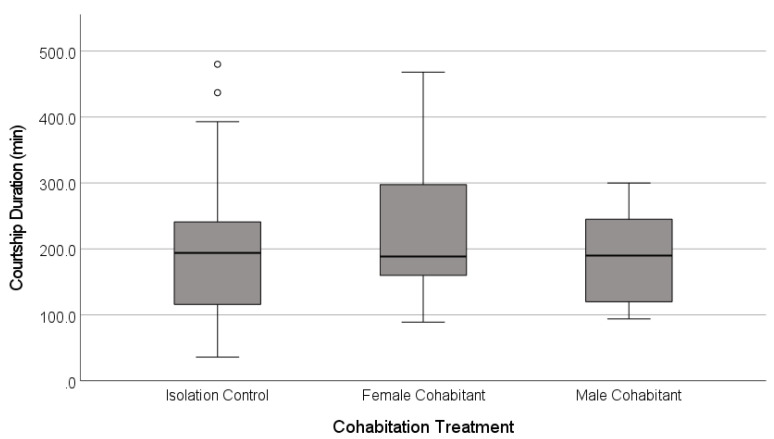
Latency to the first copulation in the laboratory mating trials for adult females that had experienced one of three social contexts during their final juvenile instar. The females were held in isolation, or were exposed to airborne, vibrational and direct tactile cues from juvenile female or adult male cohabitants. The boxplots show the medians (solid line), the range between the first and third quartiles (inter-quartile range, IQR, box), the maximum and minimum values excluding outliers (whiskers), and potential outliers (open circles, 1.5 − 3× IQR above the third quartile).

**Figure 5 insects-12-00825-f005:**
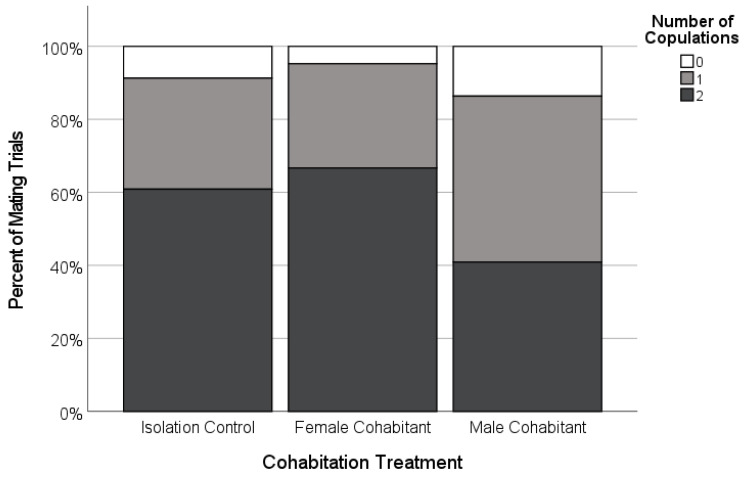
Copulation frequency of adult females paired with novel males as a function of the experimental social conditions during the female’s final subadult instar (Isolated, *n* = 23; Female-cohabitant, *n* = 21; Male-cohabitant, *n* = 22). Very few females did not copulate (white), and the proportion that copulated only once (light grey) or copulated twice (dark grey) was not significantly different across the treatments.

**Figure 6 insects-12-00825-f006:**
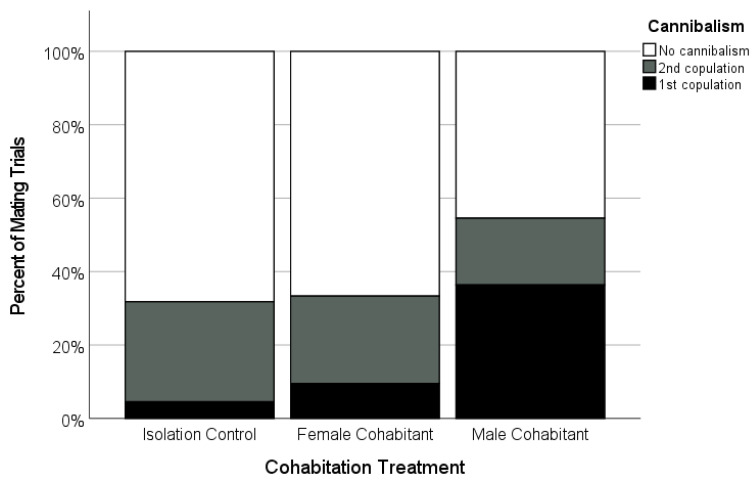
Occurrence and timing of sexual cannibalism by adult focal females that mated after being held as a subadult in one of three cohabitation treatments (Isolated, *n* = 20; Female-cohabitant, *n* = 20; Male-cohabitant, *n* = 19). Shown is the percentage of successful matings in which the females cannibalized (killed) the males during and after their first copulation (black, ‘premature cannibalism’), and during and after their second copulation (grey), as well as cases where the males survived the mating (white).

**Figure 7 insects-12-00825-f007:**
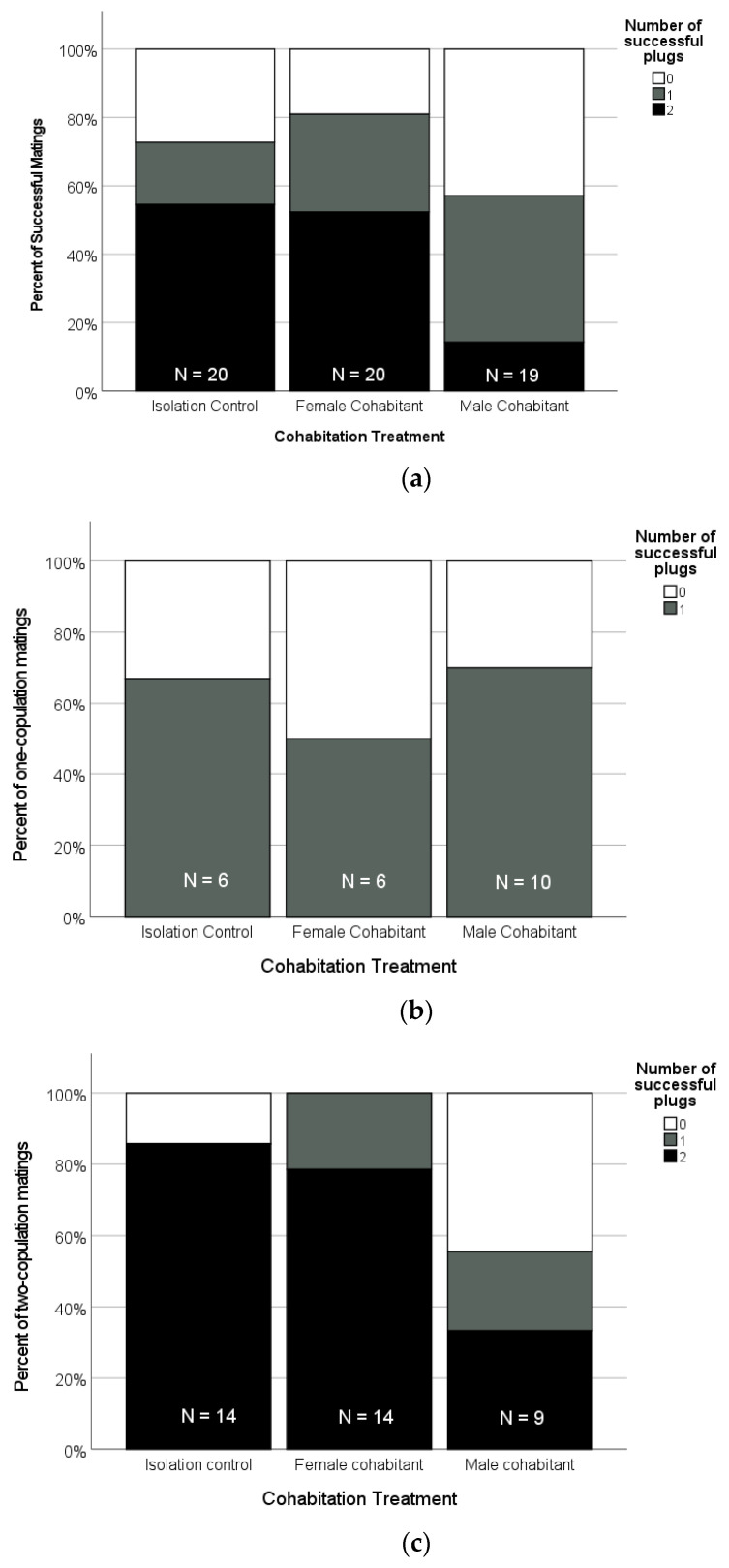
(**a**) Percent of all matings that ended with females retaining a given number of the males’ apical sclerites lodged at the entrance to the spermatheca in a position likely to block insemination by subsequent males (0 [white], 1 [grey], or 2 [black] successful plugs). (**b**) The frequency with which different numbers of successful plugs were found after the subset of matings in which the males achieved only one copulation. (**c**) The frequency with which different numbers of successful plugs were found after the subset of matings in which the males achieved two copulations. In all panels, the total sample size for each treatment is shown just above the *X*-axis.

**Table 1 insects-12-00825-t001:** Outcome of the generalized linear model tests of the effects of the male density and diet during development for adult females that copulated in mating trials in terms of latency to copulation (A), copulation frequency (B) and plug placement (C).

Model Variables	Wald χ^2^	Estimates ß (s.e.)	Model (Error Distribution)
A. Response variable: Latency to first copulation			
Density * Diet Density × Diet *	4.51, df = 1, *p* = 0.034 0.46, df = 1, *p* = 0.500 7.95, df = 1, *p* = 0.005	−0.43 (0.20) −0.15 (0.22) 0.86 (0.30)	Likelihood ratio χ^2^ = 10.39, df = 3, *p* = 0.016 (gamma)
B. Response variable: Copulation Frequency			
Density * Diet Density × Diet	4.5, df = 1, *p* = 0.034 1.01, df = 1, *p* = 0.316 0.03, df = 1, *p* = 0.863	−2.23 (1.05) −0.947 (0.944) 0.282 (1.63)	Likelihood ratio χ^2^ = 8.74, df = 3, *p* = 0.033 (ordinal logistic)
C. Response variable: Successful Sperm plugs			
Density Diet Density × Diet	2.28, df = 1, *p* = 0.131 2.81, df = 1, *p* = 0.093 1.92, df = 1, *p* = 0.166	−1.49 (0.99) −1.75 (1.04) 1.98 (1.43)	Likelihood ratio χ^2^ = 3.73, df = 3, *p* = 0.292 (ordinal logistic)

* Significant at *p* < 0.05.

**Table 2 insects-12-00825-t002:** Results of generalized linear model analyses examining the effects of the focal females’ cohabitation treatment on the outcome of successful mating trials, including the latency to copulation (A), the copulation frequency (B), the occurrence of premature cannibalism that terminated the mating (C), and the frequency of successful plug placement (D).

Model Variables	Wald χ^2^	Estimates ß (s.e.) *	Model (Error Distribution)
A. Latency to first copulation			
Treatment Male mass	1.72, df = 2, *p* = 0.424 3.32, df = 1, *p* = 0.068	-	Likelihood ratio χ^2^ = 5.13, df = 3, *p* = 0.442, (gamma)
B. Copulation frequency			
Treatment Male mass	2.37, df = 2, *p* = 0.306 0.260, df = 1, *p* = 0.610	-	Likelihood ratio χ^2^ = 2.69, df = 3, *p* = 0.442, (gamma)
C. Premature Cannibalism			
Treatment ^¥^ Male mass	8.22, df = 2, *p* = 0.016 0.235, df = 1, *p* = 0.628	Male cohabitant ^¥^: −2.64 (1.13) Female cohabitant: −0.754 (1.27) Isolated (reference)	Likelihood ratio χ^2^ = 10.19, df = 3, *p* = 0.017, (binary logistic)
D. Effectively Placed Plugs (all pairs that mated)			
Treatment ^¥^ Male mass	7.42, df = 2, *p* = 0.024 0.15, df = 1, *p* = 0.703	Male cohabitant ^¥^: −1.50 (0.631) Female cohabitant: −0.068 (0.628) Isolated (reference)	Likelihood ratio χ^2^ = 8.16, df = 3, *p* = 0.043, (ordinal logistic)

* Parameter estimates are not given if the model was not significant, ^¥^ significant at *p* < 0.05.

## Data Availability

The data analyzed in this study are available in the Dryad data repository, at https://doi.org/10.5061/dryad.n8pk0p2w5 (accessed on 10 September 2021).

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
