# Peer review of "Juvenile Experience with Male Cues Triggers Cryptic Choice Mechanisms in Adult Female Redback Spiders"

_insects, 2021, doi:10.3390/insects12090825_

Round 1

Reviewer 1 Report

The study investigates the social context of mating decisions of the female Australian redback spider Latrodectus hasselti. This is part of an extensive, ongoing study of sexual selection in widow spiders, where the redback is now a model species. In a series of elegant experiments, the authors examined the effect of male cues on choosiness of females. The approach was twofold: first, the effect of male airborne chemical cues during the penultimate instar of the female, and second, the presence of both chemical and vibratory cues of male presence on virgin adult females. The combined results support the hypothesis that females respond to male presence cues both during their development and in a situation resembling cohabitation. The results showed that while most females mated, they changed their post-copulatory behavior such that there was an increased potential for an additional male or males to copulate successfully (increased post-copulatory choosiness).

The experiments and are described clearly and discussed the context of the risk of remaining unmated, on the one hand, and the potential for mate choice in dense populations typical of many L. hasselti populations, on the other.

There are some minor points to address:

line 98: The issue is less one of seasonality and more that the spiders are short-lived (mostly annual)

lines 114-5: The sentence is incomplete. Also, it's not the delay itself that reduces survivorship, rather that survivorship declines with time (whether or not the female is mated).

lines 133-135: The first experiment included a high/low food treatment for the females, yet there is no mention here of the reason for this and no specific predictions are offered.

lines 151-152: The placement of sperm plugs – is this a rate or frequency?

Line 190: temperatures need to be corrected

line 210: cage size lacks units

lines 283-285: This is an interesting design, but it means that some females may experience more repetitions of the signals than others.

lines 303-4: correct the lens sizes

line 318: 50% ethanol? This is unusually low.

line 327: delete ‘accidents’

line 337: It’s unfortunate that cannibalism was not assessed in the airborne cues experiment.

Line 360: Halton, not Halter

line 367: In the LH treatment nearly a third of the females did not copulate (though not statistically significant). I wonder if well-fed females are willing to risk waiting longer for the right guy to come along (though this is not evident in the latencies in Fig.3).

Figure 3. Missing the letters indicating significant differences

Line 412: should be Figure 6

Figure 6: Missing the legend. Also, the rate of cannibalism seems unusually low for L. hasselti. Any explanation?

Line 460: Again, no explanation for the female diet treatment.

Reviewer 2 Report

Andrade et al. described and reported numerous mating experiments on the spider Latrodectus hasselti, in which the females were kept in different conditions.

Despite my research focus is spiders, I would like to stress that Ethology connected with sexual behaviours is far from my background, and I will limit myself in commenting the overall logic of the work as well as its results.

I find the manuscript and the work really interesting, especially the introduction in which no details are spared in describing the background. The logic of the work is also quite linear and easy to follow. Nonetheless, I have some minor comments that I think should be addressed to improve the quality of the manuscript as well as its results contextualization in spiders’ biology.

  • I would dedicate a section (or at least a paragraph) for a discussion on the hasselti population densities in the natural environment. Based on this, and starting from the fact (as the authors have clearly written) that population density may vary across time, I think it would be beneficial to discuss a little bit more about it. For example, is it more common that females are isolated (low density) or is it more common that there are sorts of islands of spiders (high density)? This is important if one wants to extend a bit these findings to spiders in general (for example there are social spiders where the trends may be the very opposite of yours, i.e. Stegodyphus). Perhaps this piece of information is contained in “Kasumovic & Andrade, un-117 unpublished”, but I would definitively add this information. Moreover, it is not impossible that the authors’ findings are related to a specific situation and may vary across time as well as the density of population does.
  • Line 292: do you surround the frame by water to avoid the spider to escape?
  • Figure 2: one copulation is enough in principle. In general, from this graph you see that the lower the density of males the lower is the possibility to have a mate. Is this not in contraction with the whole story? Stated that a single copulation may be sufficient to have a fertile cocoon, I would rather say the opposite, i.e. the more males the better, or that with more males it is likely to have at least one copulation. I guess that this must be clarified better than now or a rationale should be provided in the general statements (such as lines 23)-24.
  • Figure 3 (statistics apart): it seems that the lower density of male the less latent is the first copulation (is this not better?).
  • Figure 6 is missing a caption.
  • Figure 7: the more the plugs are successful the more likely the female should not receive further sperm. In presence of high males’ density, should the female be advantaged in getting more sperm? On the other hand, for the male is the opposite. To read this trend, it is crucial to know if it is the male that drives the insertion of the plug (I guess that it should not be the female). In this case, how these results fit with the authors hypothesis and conclusions? Can this be read in a sort of balance between the male and the female needs?
  • Line 476 there is a line missing after the references [75,85].
